# Reducing Prejudice against Children with Tungiasis: A Qualitative Study from Kenya on How a School Intervention May Raise Awareness and Change Attitudes towards Neglected Diseases

Åse Walle Mørkve *📵 and Mai Camilla Munkejord 📵

Department of Intercultural Studies, NLA University College, 5036 Bergen, Norway; mcmu@hvl.no
* Correspondence: asemor@nla.no

**Abstract:** Tungiasis/jiggers, which is caused by the sand flea, tunga penetrans, is a ferocious but neglected disease that affects millions of people in more than 80 low-income countries. If left untreated, jiggers may not only cause pain and secondary infection, but also lead to disabilities, including difficulties grasping and walking, concentration problems, sleep disturbance, skin issues and, among children, exclusion from school due to stigma and prejudice. This study aims to shed light on how a sensitising school intervention may increase awareness and improve attitudes towards jiggers among non-affected pupils. The intervention included 11 h of teaching, involving 75 pupils. In addition to teaching and observation, a pretest and a posttest were conducted. The thematic analysis of the pretest statements resulted in the following themes: "Those affected by jiggers lack knowledge", "Those affected by jiggers lack proper hygiene" and "Those affected by jiggers should be isolated from society". Moreover, thematic analysis of the posttest statements resulted in the following themes: "Increased knowledge: 'Now, I can even educate people about jiggers!'" and "Increased compassion: 'I feel bad about those people'". To foster a more inclusive school environment, including for children with disabilities due to jiggers, research on the long-term effects of similar school interventions is recommended.

**Keywords:** school intervention; stigma; prejudice; inclusive education; jiggers; tungiasis; tunga penetrans; low-income countries





## 1. Introduction

According to the United Nations (UN), everybody has the right to participate in a safe and inclusive learning environment. Children with disabilities, however, are often excluded from this right [1]. While some are born with disabilities, others develop disabilities due to illnesses. Tungiasis, a parasitic skin disease, that is caused by a sand flea called tunga penetrans is an example of an illness that may cause disabilities if left untreated (Figure 1). The parasite has common names in different countries, and is often called jiggers in English. The flea causes the disease by penetrating the skin of the host. The penetrants first appear as small black dots on the skin, causing inflammation, itching and pain. Without proper treatment, the flea can lead to secondary bacterial infection, causing chronic pain and itching, wounds, inflammation, loss of nails, lymphedema, suppuration, deep fissures and ulcers [2]. With good preventive measures and proper treatment, the risk of jiggers causing a chronic disability is limited [3]. However, prevention and treatment are often neither available nor affordable among vulnerable groups. In exposed areas, children on average get 15 new jigger penetrants every week [4]. Over time, untreated jiggers may infect hands and feet and cause difficulties walking and grasping [5], as well as disturbance of sleep and concentration [4].

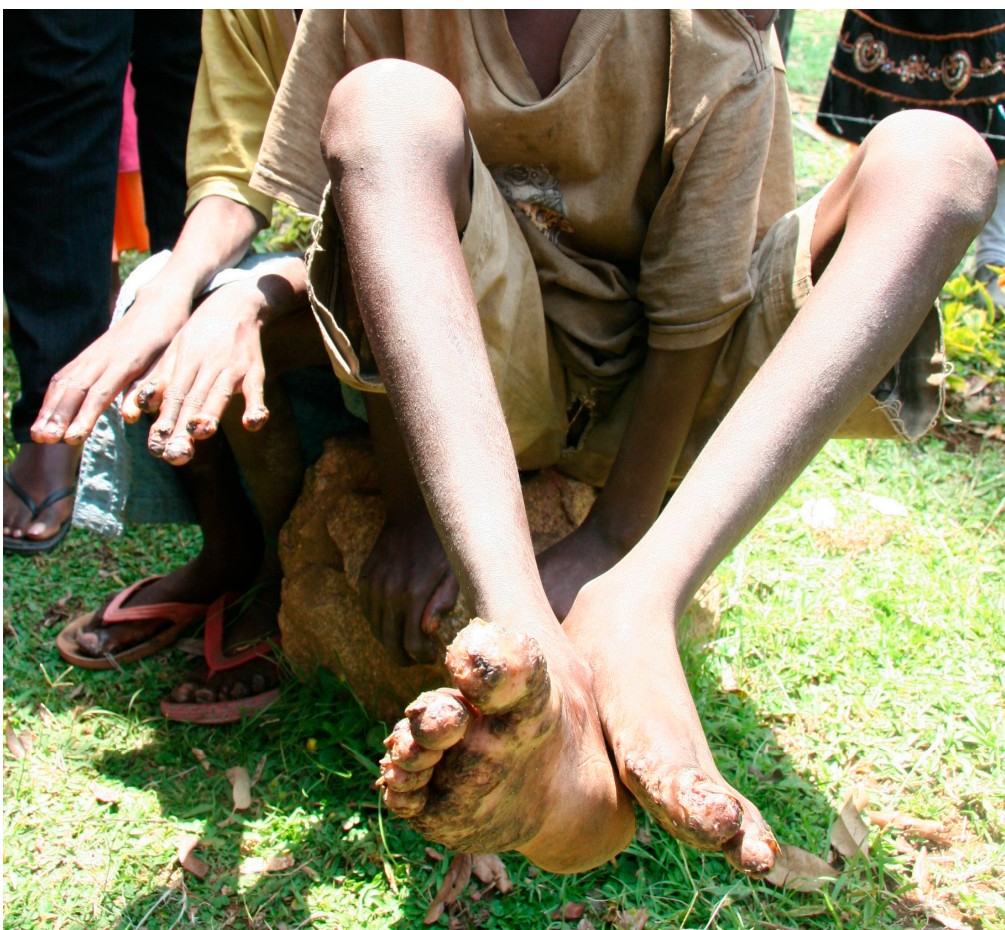

**Figure 1.** Children affected with jiggers. Picture taken by first author with permission.

Jiggers primarily affects the most vulnerable population in low-income countries, particularly in Central and South America, India and sub-Saharan countries [6]. In Kenya, where this study took place, the Ministry of Health (MOH) has published national guidelines for the prevention and treatment of jiggers [7]. By recognising jiggers as a neglected public health issue, this report identifies disability and poverty as important risk factors, as well as age, with both children and older people being more at risk than adults. Other risk factors are poor sanitation, poor hygiene, walking without shoes, and living in houses with sand floors, especially when co-habiting with animals [7,8]: factors that are, in one way or another, linked to poverty [9]. The worldwide prevalence of jiggers has never been assessed [10]. In Kenya, it is estimated that 1.4 million people, or 4% of the total population, are suffering from jiggers [7]. Impoverished communities in Kenya have documented a prevalence rate of up to 57% among children [11].

Research reveals that pupils with various types of disabilities are more likely to be stigmatised at school [12]. This has been confirmed in previous research on children with untreated jiggers, who are reported to experience bullying and social marginalisation, [10,13,14]. Moreover, there is a significant relationship between jiggers infestation and poor performance and drop out from school due to the difficulties of walking, lack of concentration and discrimination from classmates [15]. In fact, according to the MOH, in Kenya's Central Province, 50% of infected children do not attend classes at all [13]. Hence, on a societal level, jiggers disease prevents access to education and contributes to upholding poverty and marginalisation among already vulnerable populations [8,16].

In other words, jiggers disease constitutes a stigma. According to Erving Goffman, a stigma may be defined as 'the situation of the individual who is disqualified from full social acceptance' [17]. Stigma leads to persons with jiggers being socially excluded from

social events and, by being taunted by both children and adults [18], and often perceived as lazy [13]. Those suffering from jiggers are stigmatised, due not only to the skin deformities or physical disabilities caused by the disease but also due to prejudice, which contributes to denigrating them into a despised social group [17,18].

Jiggers is a ferocious, but neglected disease [10] and immediate attention is needed from the health sector and global medical industry, politicians and researchers, to reduce the suffering of those affected [7,19]. While it is of great importance to make prevention and treatment measures available and affordable, it is also vital to reduce the stigma towards children already living with the long-term consequences of jiggers. In this regard, we need increased knowledge about how to positively alter attitudes towards children affected by jiggers. Simultaneously, we need research on school interventions aimed at increasing disability awareness [20], especially in low- and middle-income countries [21], such as Kenya. This study aims to address these knowledge gaps by answering the following research question: How can a sensitising educational intervention programme increase awareness and improve attitudes towards jiggers among non-affected pupils?

In this article, we share findings from a school intervention, where the aim was to reduce stigma and to shape a more inclusive learning environment for children with skin deformities or physical disabilities caused by jiggers. Similar interventions can easily be implemented in other schools located in areas affected by jiggers disease.

## 2. Materials and Methods

### 2.1. Study Context

The study was carried out in Bungoma County in Kenya. Bungoma is located in the Western part of Kenya. In Bungoma, the population is almost 1.7 million [22]. The prevalence of jiggers is not known in Bungoma County, but is assumed to be high [16].

Participants

The study was conducted at a Private Secondary School, located in an urban and relatively affluent part of Bungoma. This school was selected because of access through the personal network of one of the members in the intervention team. The pupils came from middle- and upper-class families in Bungoma. The high prevalence of jiggers is mainly an issue for people living in rural and poor communities [18], and the pupils in this school were not expected to be highly infected by jiggers nor to know much about jiggers, which was confirmed by both teachers and the principal of the Private School. Seventy-five pupils, consisting of both boys and girls, aged 13 and 14 years of age (yoa), were included in the study.

### 2.2. Data Analysis

According to Goffman, who is considered one of the main proponents of ethnographic methods, a good way to learn is to submerge oneself in the company of the participants of a given context [17,23]. Thus, data collection for this study included participant observation in three classes, and findings collected through pre- and post-intervention tests.

### 2.2.1. The Intervention

The intervention consisted of one hour of observation and ten hours of teaching in each of the classes. The aim of the observation was to get to know the study setting. The intervention was conducted by the first author and a colleague of hers. The teaching was conducted in accordance with Lev Semenovich Vygotsky's theory of sociocultural learning. According to Vygotsky, learning can take place when individual learning (e.g., listening to a teacher) is secured by social learning (e.g., discussing, being physically active or playing with peers), and when the learners' pre-understandings (e.g., prejudice about jiggers) are brought into light and broadened [24].

Before implementing the intervention, a pretest was carried out, in which the pupils were asked to share their ideas about jiggers. The aim was to identify what the pupils knew

about jiggers, as well as their attitude towards people with jiggers disease [22]. The pupils were asked to answer one question: 'When you hear the word "jiggers", what are your thoughts?' The researchers encouraged the pupils to write down factual information they knew about jiggers, reflections on whether jiggers was an issue in their community, as well as their thoughts on jiggers disease in general. The pupils were assured that their answers would be treated anonymously. Each pupil received a piece of paper to note down their answer anonymously, while sitting by their desk. When finished, they folded the sheet and handed it in to the teacher/researcher. Our aim, when asking such an open pretest question, was to seek information on what the pupils actually knew before the intervention started, rather than focusing on what they did not know [22]. After the pretest, the pupils were informed about the upcoming program for the intervention.

During the following week, the pupils were introduced to factual knowledge on jiggers disease, including what causes jiggers, how to prevent jiggers, the consequences and symptoms of untreated jiggers, and how to treat jiggers. After these educational sessions, a person previously affected with jiggers, and now working with jiggers prevention, shared his personal story. Then, the pupils had two hours of group work, in which ten posters with questions were put up in different places in the school yard. The aim of the group work was to encourage the pupils to engage with the knowledge they had acquired during the intervention, sharing and discussing with their classmates. For example, they were asked: 'What causes jiggers?' or 'What are the symptoms of untreated jiggers?' The pupils were also asked to draw a painting including their thoughts on consequences of jiggers infection. When the intervention was completed, a posttest was carried out, asking the pupils the same single question: 'When you hear the word "jiggers", what are your thoughts?' The same procedure as for the pretest was followed.

### 2.2.2. Thematic Analysis of the Data

In this article, the author team analysed the data from the pre- and posttest to explore whether and how the educational intervention altered the pupils' knowledge about and attitudes to jiggers disease. We started by grouping the pretest answers into two piles: (a) 'stigmatising statements', i.e., answers that revealed a lack of knowledge about jiggers and/or negative attitudes to persons with jiggers vs. (b) 'non-stigmatising statements', i.e., answers that revealed correct knowledge about jiggers and/or neutral or positive attitudes to persons with jiggers. We then did the same for the posttest answers.

For example, the statements 'Those who have jiggers don't care about being clean' or 'They don't want to wear shoes' were categorised as stigmatising, whereas 'They can't afford soap' and 'They can't afford shoes' were categorised as non-stigmatising statements. Interestingly, we found 71 stigmatising pretest statements vs. 0 stigmatising statements in the posttest responses. We then performed a thematic analysis of all the pretest and posttest statements, identifying recurrent themes [22], which will be presented in the results. To avoid bias, the first author discussed the results with the colleague who was involved conducting the intervention. The colleague confirmed that she found the analysis meaningful and pertinent.

### 3. Results

The thematic analysis of the pretest statements resulted in the following themes: 1.1 Those affected by jiggers lack knowledge, 1.2 Those affected by jiggers lack proper hygiene and 1.3 Those affected by jiggers should be isolated from society. Meanwhile, thematic analysis of the posttest statements resulted in the following themes: 2.1 Increased knowledge: 'Now, I can even educate people about jiggers!' and 2.2 Increased compassion: 'I feel bad about those people'. Even though the pupils were given the option to deliver blank sheets during both the pretest and posttest, all 75 pupils chose to deliver a reply at both tests.

*3.1. Pretest Statements*

3.1.1. Those Affected by Jiggers Lack Knowledge

During the pretest, several of the pupils suggested that those affected by jiggers were ignorant. For instance, one pupil stated:

*Those that have jiggers don't have knowledge.*

Several of the pupils' statements in the pretest indicated, however, that the pupils themselves lacked knowledge on the issue of jiggers. Most were unable to write any facts about jiggers, which indicates that jiggers disease was unknown to them. Instead, most of he pupils wrote vague statements, such as:

*Jiggers is something that, if you have it, you* will *never live a good life.*

Another wrote:

*Do people with jiggers die?*

Several of the pupils were not aware that jiggers is a huge issue in Bungoma and Kenya, and a number of them wrote statements such as the following:

*Jiggers is not a disease, and it is not a big problem in Kenya.*

Moreover, a pupil simply wrote:

*I don't know.*

Rather than having factual knowledge on the issue of jiggers, the non-affected pupils shared the opinion that those affected by jiggers were themselves to be blamed.

3.1.2. Those Affected by Jiggers Lack Proper Hygiene

Several pupils stated that those affected by jiggers were themselves to blame for their situation, due to lack of proper hygiene:
One pupil wrote that:

*Those that have jiggers do not care about cleaning themselves or the house.*

On a similar note, another stated:

*They get jiggers because they do not care about maintaining proper hygiene.*

Furthermore, several of the pupils explained that jiggers would not have been an issue if those affected took precautions and properly cared for themselves:

*People in slums have jiggers because they walk barefoot. They can avoid jiggers if they are just wearing shoes.*

Furthermore, a pupil indicated that:

*Jiggers mainly attach to dirty people. If someone is dirty, they get jiggers.*

3.1.3. Those Affected by Jiggers Should Be Isolated from Society

Several of the pupils stated that they did not want to have anything to do with those affected by jiggers. For instance, one pupil wrote:

*I prefer not to play with people with jiggers.*

Another pupil wrote:

*I feel scared.*

*I want to run away.*

*I am afraid.*

Similarly, another pupil wrote:

*People get irritated with these people,*

*People are afraid of them,*

*Many run away from people with jiggers.*

*People with jiggers do not have many friends.*

Finally, one pupil stated:

*If you have jiggers in your family, others will chase you away because you might spread jiggers.*

The findings from the posttest will now be discussed.

### 3.2. Posttest Statements

3.2.1. Increased Knowledge: Now I Can Even Educate People!

While the citations from the pretest mainly consisted of short and diffuse thoughts about jiggers, in the posttest, the pupils wrote longer, descriptive sentences with correct facts about jiggers. Moreover, in most replies in the posttest, the issues of poverty and lack of hygiene equipment among people living in poverty were mentioned. For instance, one pupil elaborated that:

*I have learned more, and now I can even educate people on the importance of preventing jiggers. Most that are affected with jiggers are these people who live in poor conditions who cannot afford water and soap. These people who live in the villages might use needles which are not clean, to remove the jiggers flea.*

Another pupil wrote that:

*Jiggers are small female parasites that suck blood from human beings and animals. They also cause harm to the body. Before, I thought that jiggers were only a small issue. Now I know more.*

The pupils had understood that poverty was a risk factor for being infected by jiggers, as noted in the following citation:

*Jiggers are found in dirty places. And people living in poor conditions are in danger of getting jiggers.*

On a similar note, the pupils had understood that jiggers disease was a big problem in the society, as indicated in the following;

*It's a really big issue affecting vulnerable people . . . The children, elderly and the disabled are in danger of getting infected with jiggers.*

On several occasions during the intervention period, the author team observed that the pupils changed their perceptions towards jiggers infection. Looking at the pupils' drawings and the replies on the group work indeed supported this observation, where the pupils were showing compassion towards those infected. The pupils themselves also emphasised that they had changed their mind. They explained that they used to think that those with jiggers were ignorant. But now they knew better and would try to help if they met someone suffering from jiggers. One pupil wrote in the posttest:

*I have learned more, and now I can even educate people on the importance of preventing jiggers. . . Jiggers is a real disease, and we should work together to stop it.*

Another pupil noted:

*I thought that jiggers were harmless, but now I know better.*

Many of the answers during the posttest, in other words, indicate that the students had acquired a lot of factual knowledge about the disease, jiggers. In addition, the answers provided by the pupils clearly showed that they had understood the negative social effects of stigmatisation for those affected, which will now be discussed.

### 3.2.2. Increased Compassion: I Feel Bad about Those People

During the intervention period, we observed a growing compassion for those infected by jiggers disease. Our observations from the classroom indicate that listening to a person affected by jiggers telling his personal story was particularly revealing for the pupils. In addition to talking about challenges related to poverty and vulnerability, this person showed pictures of people suffering from jiggers. Also, the pupils were challenged to reflect on various dilemmas during the intervention period, for example: 'If you didn't have money to buy enough food for your children, would you prioritise buying soap to wash yourself?' and 'How do you think those affected feel when people laugh at their condition?' This encounter with a person living with jiggers and reflecting on the dilemmas and challenges he faced in his everyday life resulted in compassion for those affected by jiggers. This transformation was also observed from the answers in the posttest. For instance, one pupil stated:

> *I think people with jiggers are not well taken care of.*

Another pupil described how his or her own attitude towards people with jiggers had changed as a result of the intervention:

> *I didn't know that jiggers were painful. Before, I used to laugh at those that had jiggers, even though they were crying . . . I know now that I was wrong.*

Another pupil noted:

> *People with jiggers must feel lonely . . . It's not good to isolate the people suffering from jiggers.*

Furthermore, another pupil stated:

> *I feel bad about those people who have jiggers, and someday, when I grow up, I want to be part of treating it.*

Finally, a pupil stated in the posttest:

> *I never imagined that jiggers could cause so much harm to the country.*

## 4. Discussion

When the main author and her colleague arrived at the school in order to implement this intervention, some of the local teachers laughed at the fact that the intervention focused on jiggers disease, and the principal stated that he did not consider jiggers an important topic to teach the pupils about. The fact that jiggers is often a neglected topic in the school curriculum is confirmed in previous research [16]. However, after an open discussion with the principle and the staff, our teaching plan was accepted. As indicated in our findings, our analysis of the pretest statements indicated that the pupils included in this study thought that those affected by jiggers lacked knowledge, did not maintain proper hygiene and should be isolated from society.

Research, however, indicates that poverty is the main risk factor for jiggers infection, as those at risk often lack access to clean water, soap, shoes, and solid floors in the homestead, and, when infected, they may lack the necessary means to pay for treatment [25,26]. However, in the pretest, none of the pupils mentioned the issue of poverty. Rather, they stated that those affected were dirty or did not care to wear shoes, as if blaming the poor for their living conditions. Moreover, our pretest found a relationship between stigma and social interaction as also indicated in previous research [27]. In our study, pupils stated that they preferred to maintain a distance from those affected by jiggers, e.g., not playing with them or staying away from them in the classroom. Most of the pupils preferred to stay away from those affected, to keep themselves safe from this disease, as they thought that the disease could spread from person to personand they were afraid of being contaminated. Inspired by Goffman, we can say that, due to the imagined danger they represent according to those not affected, persons with jiggers are marked with shame. And as a consequence, suffer from stigma or what we may call a spoiled social image [17].

After a few hours of teaching, discussions, and reflections during our intervention, it was observed that the pupils gradually changed their mindset about jiggers, both by increasing their factual knowledge and by increased empathy vis-à-vis those affected. According to our knowledge, similar intervention studies aiming to reduce stigma towards people suffering from jiggers have not been evaluated in the scientific community. However, previous research on stigma towards people with tuberculosis found that community-based education interventions seemed to significantly decrease the level of stigmatizing attitudes [28]. Similarly, several school interventions have successfully contributed to reducing stigma towards pupils with HIV or leprosy [29–31]. Even short-term interventions can be quite effective in altering attitudes and increasing knowledge, according to a systematic review of ten school-based sexual health interventions of similar constitution to the intervention described in this study [31].

Similarly, in the posttest of our intervention, the pupils revealed increased knowledge and a broader understanding about jiggers, as well as increased compassion for those affected, as we saw during the group work and the pupils drawings during the intervention. In fact, those with factual knowledge about a disease are less likely to have pejorative attitudes or act in a discriminatory way towards persons affected by the disease [32,33]. As previously mentioned, stigmatising of vulnerable groups constitutes a major obstacle and contributes to excluding people from education, paid work and a good social life [17]. Pupils infected with jiggers are at high risk of school drop out [10,13] attend class less frequently and have poor school performance [15]. In addition, stigmatization of neglected diseases such as jiggers, might contributes to poor health seeking behavior [16]. Thus, sensitising interventions are crucial if we are to be able to stop the ongoing exclusion and marginalisation processes against persons with illnesses and various forms of disabilities [21,34].

## 5. Limitations and Concluding Remarks

A potential limitation in the study might be that, in the pretest, the pupils did not know what the teacher expected them to write and, in the posttest, some of them may have provided socially desirable answers, rather than their own thoughts. Instead of conducting the intervention at a Private Secondary school, it could have been even more useful to conduct a similar intervention program at a Primary Public school, where those pupils most at risk for jiggers infection are studying. Furthermore, we do not know how long the promising changes identified in this study lasted, not do we know if the increased knowledge and improved attitudes resulted in more inclusive behaviours outside the classroom. Adding a follow-up posttest 6 months after the intervention, would have provided information about the longer-term effects of the intervention [35]. Probably, this kind of intervention need to be done on a regular basis to have a bigger impact on the practices of pupils, teachers and the community at large. While we acknowledge that experiential learning is a powerful pedagogical tool, and that school interventions are indeed cost-effective and resource-friendly, we simultaneously emphasise the need for further studies on the long-term effects of such interventions, and the importance of doing such interventions on a regular basis. Previous research indicates that particularly promising results may be obtained when combining in-class teaching with a community-based component [36]. However, more research is needed on how anti-stigmatising and knowledge-raising school interventions can be carried out as efficiently as possible, in order to develop a more inclusive educational environment for all, including for children with various forms of conditions and disabilities [21,28]. Our hope is that our study may constitute one step in the right direction in this regard.

**Author Contributions:** Conceptualization, Å.W.M., M.C.M.; methodology, Å.W.M., M.C.M.; analysis, Å.W.M., M.C.M.; investigation, Å.W.M.; data curation, Å.W.M., M.C.M.; writing—original draft preparation, Å.W.M.; writing—review and editing, Å.W.M., M.C.M.; visualization, Å.W.M., M.C.M.; supervision, M.C.M.; project administration, Å.W.M. All authors have read and agreed to the published version of the manuscript.

**Funding:** This research received no external funding.

**Institutional Review Board Statement:** The study was conducted in accordance with the Declaration of Helsinki. The intervention as such was developed and conducted in collaboration with staff at the Department of Public Health and the Department of Education at Moi University, Kenya. As no personal information was gathered/registered about any of the participants, NSD—Norwegian Centre for Research Data concluded that the study could be done without ethical approval.

**Informed Consent Statement:** Informed consent was obtained from the principal who invited us to do this intervention in their school. As for the pupils themselves, they had to take part in the various parts of this intervention, however they did not have to answer our question in the pretest and posttest. They provided anonymous answers, and even blank sheets would have been accepted.

**Data Availability Statement:** As the data consist of handwritten, curved paper from pupils, the data has not been made available.

**Acknowledgments:** We want to express our gratitude to the staff of the Private School. Thank you to staff and volunteers of Bungoma Red Cross, for facilitating our stay. To Linda Natås who took part in the data collection; thank you for your cooperation and useful conversations throughout our stay in Bungoma. Finally, to the members of our research group "Intercultural Studies, inclusion and social justice"; thank you for input and support to earlier drafts of this article.

**Conflicts of Interest:** The authors declare no conflict of interest.

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
