# Peer review of "Reducing Prejudice against Children with Tungiasis: A Qualitative Study from Kenya on How a School Intervention May Raise Awareness and Change Attitudes towards Neglected Diseases"

_societies, doi:10.3390/soc13060139_

Round 1

Reviewer 1 Report

Jiggers is a dangerous disease caused by the sand flea. It is little known and rare in the European population.

The summary briefly but correctly and sufficiently informs about the school intervention and its results against Tungiasis/jiggers.

Keywords contain references to all aspects considered in the article.

In the introduction, the authors explain that Tungosis, a parasitic skin disease, can be disabling if left untreated. Prevention and treatment are often not available. Untreated jiggers can cause other disorders and disabilities.

In the introductory part, there is also a photo of a child with jiggers. In the introduction, the risk factors and epidemiology of the disease are also given, and the exposure of sick children to stigmatization by healthy people in the school environment and local community is marked.

This study aimed to increase knowledge about how to positively change attitudes towards children affected by jiggers.

In this article, the authors share the results of a school intervention on how to shape a more inclusive learning environment for children with physical disabilities caused by jiggers. The Authors suggest that similar interventions may also be implemented in other schools located in areas affected by jiggers disease.

The study was conducted among 75 students aged 13 and 14. They were preceded by a pretest, and after the tests, a posttest was performed. The ethnographic method was used in the research. In this article, the authors analyzed data from the pre-test and post-test to investigate whether and how the educational intervention changed students' knowledge about attitudes towards jiggers disease. This process is described in detail in the paper.

In the discussion, the Authors noted the lack of intervention studies aimed at reducing the stigmatization of people with jiggers in the scientific community. Therefore, the Authors compared their results to a study on the stigmatization of people with tuberculosis, HIV and leprosy. In the limitations of the study, the authors rightly pointed out the need to add a follow-up post-test 6 months after the intervention. All literature items have been referred to in the text of the work. This Authors study is the first to assess the knowledge and stigmatization of people with jiggers.

I highly appreciate this research in terms of its content and innovation, and I recommend it for publication.

Author Response

Thank you very much for reviewing our article. Your comments are highly appreciated.  

Please find attached a document, with a point-by-point response to the reviewer’s comments.

Reviewer 2 Report

Summary

This manuscript describes the outcome of an intervention to reduce stigma and discrimination towards people with jiggers. The study uses qualitative methods to assess the attitudes of teenagers in a private secondary school in an endemic area, but who are not infected themselves, before and after the intervention. The intervention is important and much needed in endemic areas since stigma prevents people infected with jiggers from participating in normal activities and seeking help. This study is the first to attempt to do this and demonstrates that a simple intervention in a school can change the attitudes of teenagers.

Introduction

Throughout the description of the disease and parasite are not quite accurate and sentences in lines 29, 34, 36, 37, 41 should be edited.

 The disease is called tungiasis which is caused by the parasite, Tunga penetrans.  The parasite has common names in different countries including sand flea and in East Africa, jiggers.

 The flea causes the disease by penetrating the skin of mammalian hosts and inducing inflammation which causes extreme pain and itching. This can be aggravated by secondary bacterial infection.

There is some discussion regarding its status as an infestation or infection, but the WHO classification is as an infection so this word should be used.

 Tungiasis is an infection immediately a single flea penetrates the skin, it is not “without proper treatment, the flea can lead to infection”.

 Line 36 Blood poisoning is not a scientific term. I presume the authors are referring to secondary bacterial infection which first causes abscesses and suppuration.

 Line 37. The fleas mostly penetrate the feet and occasionally other body parts, not the legs.

 Line 41 Referring to “untreated disease” is not appropriate since almost all patients treat themselves by attempting to extract the fleas.

 Line 46 “children and older people being more exposed than adults” They are more at risk or the prevalence is higher in these age groups.

Line 57. The citation is not a peer reviewed manuscript and is not a strong study to report as anything other than anecdotal evidence.

 Line 67 it is not usual to include a page number in a citation.

 Line 72. No need to summarize the introduction.

 Methods

The aims of the study are described appropriately in the Introduction so shouldn’t be repeated in Methods. In addition, there is an aim in line 90 for which no results are presented nor discussed.

 Methods should start with a description of the study population/participants and location. Include the prevalence of tungiasis in the area if available. The sex of the participants should also be included.

 There should be a description of the qualifications and training of the team who conducted the intervention and the evaluation. Was it the same people for all aspects?

 The name of the school should never be mentioned. This is a breach of confidentiality of the participants and code of ethics. It is possible to locate a private secondary school of this name in the described area on Google maps. The name of the school does not need to be mentioned at all.

 Data collection should be in the section with Data Analysis.

 It would be appropriate and helpful for readers and researchers wishing to replicate the work, to include the intervention curriculum and materials in the supplementary materials.

 The format of the pre- and post-test needs more detail. Were these conducted in private or in a public exam type format? Make it clear there was only the one question asked.

 What were the informal conversations? Was there a guide for these? Were they recorded? Which pieces of data came from those? If they provided no data/information they should not be mentioned in the methods.

 What was the “one hour observation”? Was that part of the pre-test or the intervention itself? Was there an observations guide? How was the data used?

 Were any of the preliminary conversations, observations or the pre-test used to inform the development of the intervention?

 Was there an SOP or guide for the session with the affected person?

 Line 127. What is meant by “pupils draw their thoughts”? Were they asked to draw a picture of their thoughts? What was the objective of this component? Where are the results of this?

 Line 141. Figure 2 is actually a table, but it is unnecessary since the little information contained is well described in the text.

Lines 154 & 157 are results and should be moved to that section.

 Data analysis: who conducted this? Was it done by 2 individuals separately and a third checked any disagreements in themes? How did you avoid bias?

 Results

It would help to have an introductory paragraph describing how the results are presented/ structured. Include in this, the sentences from the methods line 154 and 157.

 It would be appropriate to indicate what proportion of the 75 pupils gave responses in the pre- and post-tests.

What were the findings of the observations and casual conversations and the drawings?

 Line 172. It is not appropriate to describe what the pupils did not write in the results section. You could mention this in the discussion.

 Line 184. There are not chapters in a manuscript. Delete this.

 Line 198. The word “Finally” is used and yet this is not the final result. Delete the word.

 Line 200. This is not a result. Should be moved to the discussion. However, there is no evidence that shoes are protective. One prevention trial has shown that they do not work:

(Thielecke M, Raharimanga V, Rogier C, Stauss-Grabo M, Richard V, et al. (2013) Prevention of Tungiasis and Tungiasis-Associated Morbidity Using the Plant-Based Repellent Zanzarin: A Randomized, Controlled Field Study in Rural Madagascar. PLoS Negl Trop Dis 7(9): e2426. doi:10.1371/journal.pntd.0002426)

 Line 220. It is not necessary to summarise results immediately after presenting them. Maybe appropriate at the beginning of the discussion.

 Line 221. “this was an issue rarely discussed in affluent homes” what piece of evidence have you presented to support this statement in summarizing the pre-test results.

 Line 247. The word “weakest” is not appropriate to be used to describe people in a manuscript.

 Discussion

Line 298. Poverty is the main risk factor for jiggers, not “the underlying cause of the prevalence of (untreated) jiggers infestation”. Please change this sentence. Make it clearer that it is the poverty that causes a family to “lack access to clean water, soap, shoes, and solid floors in the homestead….” which puts them at risk for jigger infection.

 Line 326-328. The lack of interventions to reduce stigma to people with HIV does not have any relevance to the manuscript.

 The authors should double check all of the references to ensure they are correctly cited and correctly written in the reference list. For example:

Line 332. The cited paper is about TB not NTDs.

Line 333. The cited paper does not give evidence on school dropouts nor bullying by teachers and pupils.

Reference list

(3) J. Heukelbach, S. Franck og H. Feldmeier

(4) S. Wiese, L. Leson og H. Feldmeier,

 Limitations

Line 345. Is there any evidence in the literature that this type of intervention should be done regularly? At what interval? Would strengthen the section to have a citation.

 It would be appropriate to mention the limitation of conducting this intervention in a private school compared to a public school known to have infected pupils. Likewise, the use of a secondary school rather than a primary school where the age group most at risk for jiggers are studying.

Author Response

(The authors gave the same response as above.)
